# Linear mitochondrial DNA is rapidly degraded by components of the replication machinery

Viktoriya Peeva[1], Daniel Blei[1], Genevieve Trombly[1], Sarah Corsi[1,7], Maciej J. Szukszto[2], Pedro Rebelo-Guiomar[2,3], Payam A. Gammage[2], Alexei P. Kudin[1], Christian Becker[4], Janine Altmüller[4,5], Michal Minczuk [2], Gábor Zsurka [1,6] & Wolfram S. Kunz [1,6]

Emerging gene therapy approaches that aim to eliminate pathogenic mutations of mitochondrial DNA (mtDNA) rely on efficient degradation of linearized mtDNA, but the enzymatic machinery performing this task is presently unknown. Here, we show that, in cellular models of restriction endonuclease-induced mtDNA double-strand breaks, linear mtDNA is eliminated within hours by exonucleolytic activities. Inactivation of the mitochondrial 5′-3′ exonuclease MGME1, elimination of the 3′-5′exonuclease activity of the mitochondrial DNA polymerase POLG by introducing the p.D274A mutation, or knockdown of the mitochondrial DNA helicase TWNK leads to severe impediment of mtDNA degradation. We do not observe similar effects when inactivating other known mitochondrial nucleases (EXOG, APEX2, ENDOG, FEN1, DNA2, MRE11, or RBBP8). Our data suggest that rapid degradation of linearized mtDNA is performed by the same machinery that is responsible for mtDNA replication, thus proposing novel roles for the participating enzymes POLG, TWNK, and MGME1.

---

[1] Institute of Experimental Epileptology and Cognition Research, University of Bonn, Sigmund-Freud-Str. 25, D-53105 Bonn, Germany. [2] MRC Mitochondrial Biology Unit, University of Cambridge, Cambridge CB2 0XY, UK. [3] Graduate Program in Areas of Basic and Applied Biology (GABBA), University of Porto, Porto 4200-135, Portugal. [4] Cologne Center for Genomics, Center for Molecular Medicine Cologne (CMMC), University of Cologne, Weyertal 115b, Cologne D-50931, Germany. [5] Institute of Human Genetics, University of Cologne, Kerpener Str. 34, Cologne D-50931, Germany. [6] Department of Epileptology and Life & Brain Center, University of Bonn, Sigmund-Freud-Str. 25, Bonn D-53105, Germany. [7] Present address: Human Nutrition Research Centre, Institute of Cellular Medicine, Newcastle University, Newcastle upon Tyne NE2 4HH, UK. These authors contributed equally: Viktoriya Peeva, Daniel Blei. Correspondence and requests for materials should be addressed to M.M. (email: michal.minczuk@mrc-mbu.cam.ac.uk) or to G.Z. (email: gabor.zsurka@ukbonn.de) or to W.S.K. (email: wolfram.kunz@ukbonn.de)

The mechanism of mtDNA degradation and its role in the maintenance of the mitochondrial genome is an important unsolved question of cell biology. Due to the multicopy nature of the mitochondrial genome, the fate of damaged DNA molecules is substantially different in mitochondria in comparison to the nucleus. Damaged DNA molecules normally represent only a tiny fraction of total mtDNA of a cell, which can be discarded without severe consequences and replaced by replicating intact mtDNA. This idea of a 'disposable genome'[1, 2] plays a crucial role in emerging approaches in gene therapy of mitochondrial DNA diseases that aim to reduce the proportion of pathogenic mtDNA mutations by selectively cleaving and subsequently breaking down mutated mtDNA[3–5]. Active elimination of mtDNA has also been described as an important aspect of restricting transmission of paternal mtDNA in animals[6].

Here, we show that the main protein factors that perform rapid exonuclease-mediated degradation of linear mtDNA are well-known components of the mtDNA replication machinery. Failure to eliminate damaged mtDNA leads to accumulation of abnormal linear and rearranged molecules. Thus, we demonstrate that the removal of linear mtDNA is an important aspect of proper maintenance of the mitochondrial genome.

## Results

**Exonuclease-mediated degradation of linear mtDNA**. To investigate the degradation of linear mtDNA, we created modified HEK 293 cell lines expressing mitochondria-targeted restriction endonucleases (mtRE) under the control of the tet-on promoter. One of the cell lines expressed mitoEagI that introduced a double-strand break in mtDNA at a single site within the minor arc (scheme in Fig. 1a), while the other cell line expressed mitoPstI that cut mtDNA at two sites within the major arc (scheme in Fig. 2a). Southern blotting confirmed the efficient cleavage of the mtDNA (Fig. 1a, b and Fig. 2b). In control mitoEagI-expressing cells, linear mtDNA species carrying an end that corresponded to the original cutting site were only abundant after 2 h of induction (Fig. 1a, c, mitoEagI). At later time points, a complex mixture of smaller sized mtDNA fragments was detectable with prominent bands from a few hundred up to several thousand base pairs downstream from the cutting site (Fig. 1a). The band representing an end at 700 bps distance from the cutting site appeared at 4 h and then faded away, while the band at 3.2 kb distance reached its peak at 6 h. These observations are compatible with a degradation process that starts at free ends and then gradually progresses over several thousand nucleotides.

To obtain a more detailed picture of cleavage-dependent alterations, we performed ultra-deep sequencing of the mitochondrial genome in mitoEagI-expressing HEK 293 cells 6 h after induction and calculated mtDNA coverage ratios in induced vs. non-induced cells (Fig. 1d, gray). Decreased coverage ratios are indicative of linearization-dependent removal of parts of the mitochondrial genome. In control cells, the coverage ratio was dramatically reduced in the vicinity of the cutting site and gradually increased distal to both ends. This suggests again that the bulk degradation of linear mtDNA takes place starting from free ends, which is characteristic for the activity of exonucleases.

**Main components of the mtDNA degradation machinery**. We hypothesized that MGME1, a mitochondrial DNA exonuclease of which pathogenic mutations cause perturbation of mtDNA maintenance and multisystemic mitochondrial disease[7, 8], could be at least partly responsible for the exonucleolytic activity observed upon mtRE-induced DNA double-strand breaks in the mitochondrial genome. Therefore, we knocked out the *MGME1* gene in mitoEagI-expressing HEK 293 cells using the

CRISPR–Cas9 technology and obtained two clones with different frame-shift deletions in exon 2 (Supplementary Fig. 1). Southern blotting showed that loss of the MGME1 activity did not alter the efficiency of mtDNA cleavage (Fig. 1a, b; Supplementary Fig. 2a), but resulted in long persistence of the non-degraded mitoEagI-linearized fragment (Fig. 1a, c; Supplementary Fig. 2a). Ultra-deep sequencing showed that the coverage ratio was unaltered throughout the whole mitochondrial genome in MGME1 knockout cells 6 h after inducing DNA double-strand breaks (Fig. 1d, red; Supplementary Fig. 2b). Comparing the amounts of retained mtDNA in controls and MGME1 knockout cells (represented by sums of total normalized reads) revealed that the latter lost at least 96% of their capacity to rapidly degrade linear mtDNA. This suggests that MGME1 exonuclease plays a central role in the removal of linear mtDNA species. We did not observe alterations of MGME1 protein amounts during the investigated time course (Supplementary Fig. 3), which indicates that rapid mtDNA degradation is performed by pre-existing MGME1 protein molecules.

MGME1 has a strong preference for degrading single-stranded DNA species in 5′-3′ direction[7–9]. Therefore, we hypothesized that a second exonuclease, working in 3′-5′ direction, could be responsible for the simultaneous degradation of the opposite mtDNA strand. The mitochondrial replicative DNA polymerase POLG is known to exhibit a 3′-5′ exonuclease activity, and thus has been identified as potential candidate. Notably, POLG has been previously observed to be physically associated with MGME1[8]. We introduced the p.D274A mutation in POLG in mitoEagI-expressing HEK 293 cells using the CRISPR–Cas9 technology (Supplementary Fig. 1). Equivalent mutations have been demonstrated to lead to selective elimination of the 3′-5′ exonuclease activity of POLG in yeast[10], mice[11, 12], and flies[13]. Similarly to the *MGME1* knockout, the p.D274A POLG mutation led to a severe inhibition of mtDNA degradation as observed by Southern blotting (Fig. 1a, c) and ultra-deep sequencing (Fig. 1e, green). This suggests that the 3′-5′ exonuclease activity of POLG is also required for efficient removal of linear mtDNA species. Polymerase γ performs DNA synthesis as a heterotrimer composed of one catalytic subunit (coded by the *POLG* gene) and two copies of the accessory subunit (coded by the *POLG2* gene)[14]. In order to investigate whether the accessory subunit also plays a role in mtDNA degradation, we knocked down POLG2 by siRNA treatment. We did not observe an inhibition of mtDNA degradation upon loss of POLG2, which suggests that, in opposite to the replication function of POLG, degradation does not require the binding of the accessory subunit (Supplementary Fig. 4a).

We used an additional cellular model to verify the prolonged persistence of non-degraded, mtRE-generated linear mtDNA upon partial inactivation of MGME1 or POLG, and performed siRNA-mediated knockdown for these proteins in mitoPstI HEK 293 cells (Fig. 2). Since it has been demonstrated that MGME1 and the 3′-5′ exonuclease activity of POLG both have strong preferences for single-stranded DNA[8–10], we reasoned that the observed exonucleolytic degradation of linear double-stranded mtDNA would require the additional operation of a helicase in order to unwind the two DNA strands. To address this hypothesis, we knocked down the expression of the mitochondrial replicative DNA helicase Twinkle (TWNK) using the siRNA technique. In accordance with the central role of TWNK in mtDNA replication, knockdown resulted in decreased mtDNA copy numbers (~15% of controls, Supplementary Table 1). Southern blotting showed an increased abundance of the non-degraded mitoPstI-dependent linear 2.1-kb mtDNA fragment in TWNK knockdown cells, which was comparable to those observed in MGME1 or POLG knockdowns (Fig. 2a, c). Detecting free mtDNA ends by single-molecule amplification of

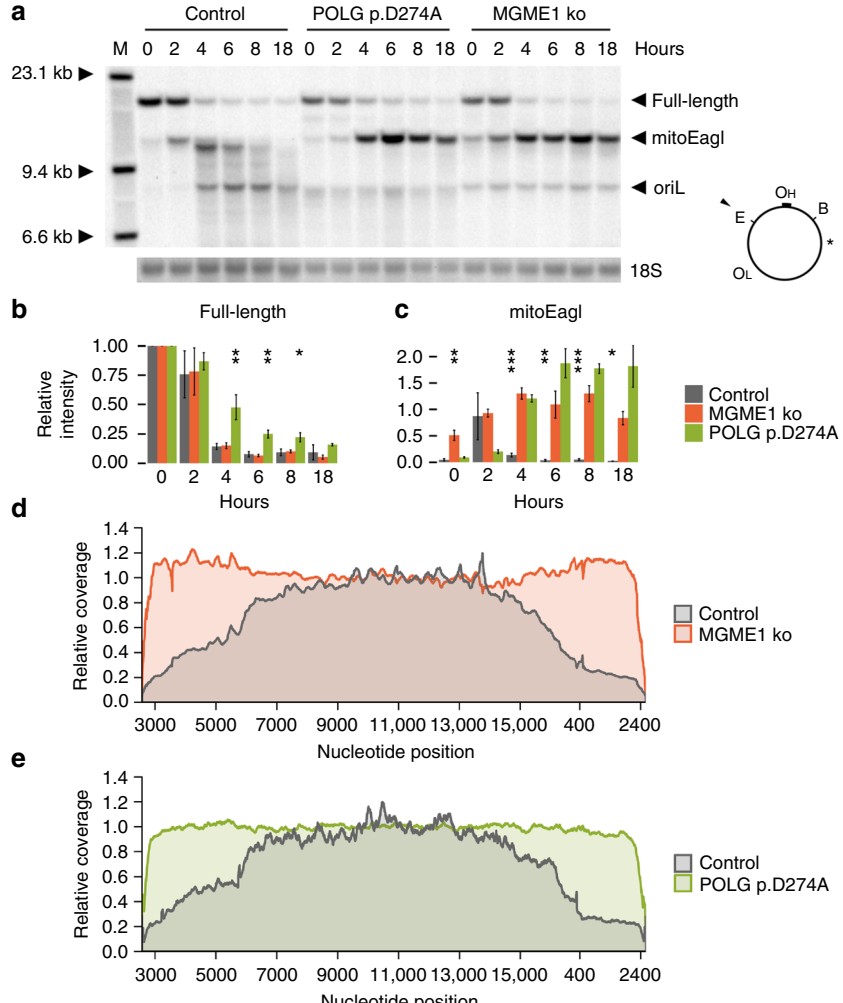

**Fig. 1** Degradation of linearized mtDNA in control and mutant mitoEagI-expressing HEK 293 cells. **a** Southern blot showing the degradation of mtDNA within the first 18 h of induced expression of mitoEagI (E) in control, MGME1 p.I9Qfs*32 knockout ('MGME1 ko'), and exonuclease-deficient POLG ('POLG p.D274A') cells. BamHI endonuclease-linearized DNA (B) was labeled with a mitochondrial probe represented by an asterisk as well as a probe specific for nuclear 18S ribosomal DNA ('18S'). Note that persistent bands with one end in the vicinity of oriL in MGME1 ko and mutated POLG cells are already present before induction (time point '0', lowest arrowhead). These linear mtDNA species are due to leaky mitoEagI expression and their presence is not related to the induced massive double-strand breaks. **b** Quantification of full-length mtDNA confirms the efficient cleavage of mtDNA by mitoEagI and (**c**) the persistence of mitoEagI-linearized mtDNA in MGME1 ko and POLG p.D274A cells. Band intensities were first normalized to 18S ribosomal DNA intensities then to intensities of the full-length mtDNA in each cell line before induction. Error bars represent standard errors of the mean (SEM) in three independent experiments (including both available MGME1 knockout clones). Significance was calculated by applying one-way ANOVA test. *$P < 0.05$, **$P < 0.01$, *** $P < 0.001$. **d**, **e** Coverage ratios throughout the mitochondrial genome as determined by ultra-deep sequencing of mtDNA from cells 6 h after induced mitoEagI expression and normalized to values in non-induced cells. In control (gray), coverage ratio is the lowest around the mitoEagI cutting site (represented by the two ends of the x-axis) and gradually increases in both directions before reaching full coverage. In MGME1 ko cells (**d**, red) and in cells with exonuclease-deficient POLG (**e**, green), coverage ratio drops only in the immediate vicinity of the mitoEagI cutting site. Note that library preparation techniques used for ultra-deep sequencing result in underrepresented positions in the close vicinity of free DNA ends

linker-ligated DNA species and subsequent Sanger sequencing (Supplementary Fig. 5a) confirmed the decreased efficiency of degradation of linear mtDNA in MGME1, POLG, and TWNK knockdown cells (Fig. 2d, e). Thus, our data indicate that MGME1 exonuclease, the exonuclease activity of POLG polymerase, and the Twinkle helicase participate in degrading linear mtDNA. We did not observe similar effects on mtDNA degradation upon complete or partial loss of other candidate nucleases such as EXOG[15], APEX2[16], ENDOG[17], FEN1[18], DNA2[19, 20], MRE11[21, 22], or RBBP8[22] (Supplementary Fig. 4). Our finding that MGME1 siRNA-mediated knockdown inhibits degradation of linear mtDNA in mitoPstI-expressing HEK 293 cells is in apparent contradiction with a recent report where no

effect of MGME1 knockdown was detected[2]. Western blot analysis revealed that the longer depletion time used in our study (6 days instead of 3 days in the previous study) leads to more efficient depletion of the MGME1 protein, which may explain the conflicting conclusions (Supplementary Fig. 6).

**Double-stranded mtDNA degradation intermediates.** To unravel the nature of degrading linear mtDNA molecules, we performed ultra-deep sequencing of linker-ligated mtDNA ends (Fig. 3a). Ligating the one-side-blunt double-stranded linker to either native or T4-polymerase-blunted mtDNA (Supplementary Fig. 5a) resulted in similar overall quantities of detected ends

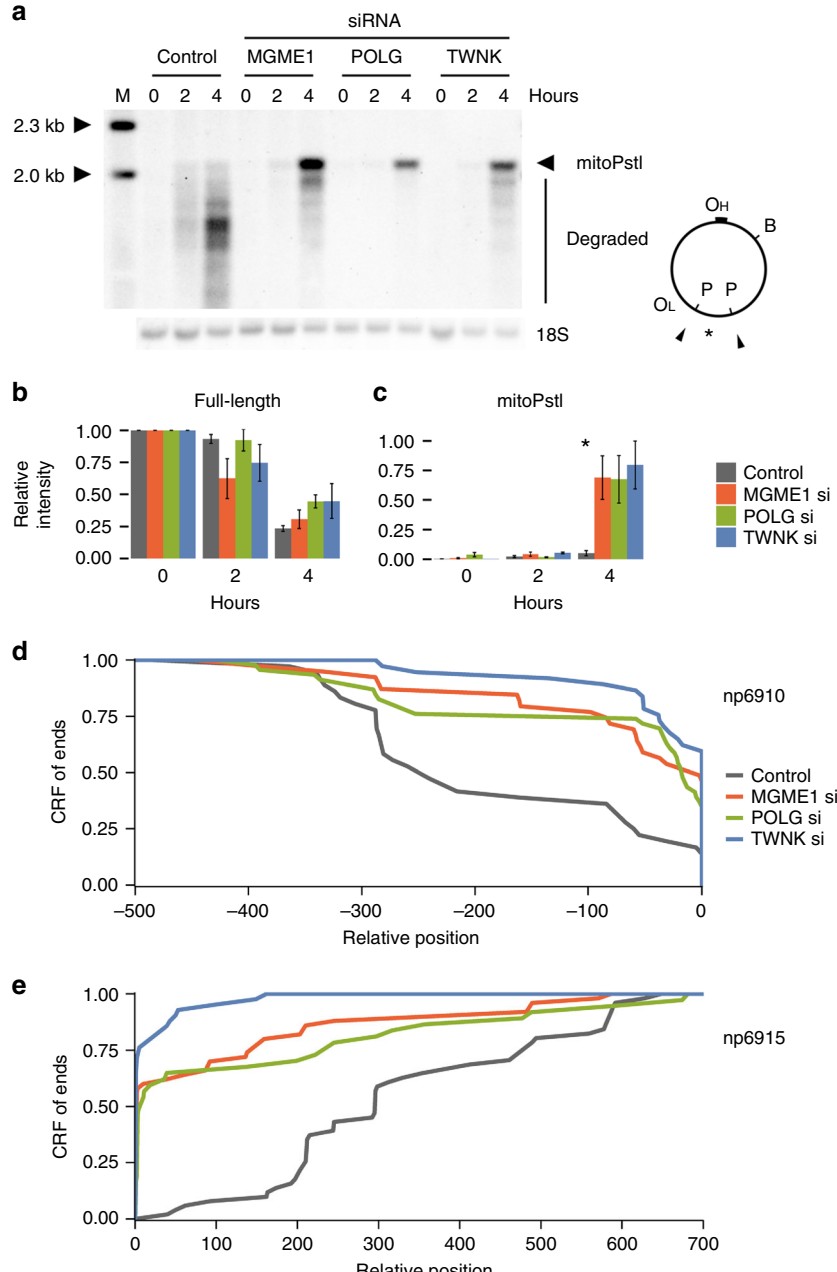

**Fig. 2** Degradation of the 2.1-kb mtDNA fragment in control and siRNA-treated mitoPstI-expressing HEK 293 cells. **a** Southern blot showing the appearance and degradation of the 2.1-kb mtDNA fragment within the first 4 h of induced mitoPstI (P) expression in control cells and in *MGME1*, *POLG*, or *TWNK* siRNA-treated HEK 293 cells. BamHI endonuclease-linearized DNA (B) was labeled with a mitochondrial probe represented by an asterisk as well as a probe specific for nuclear 18S ribosomal DNA ('18S'). **b** Quantification of full-length mtDNA confirms the efficient cleavage of mtDNA by mitoPstI and **c** the persistence of the 2.1-kb mtDNA fragment in *MGME1*, *POLG*, and *TWNK* knockdown cells. Band intensities were first normalized to 18S ribosomal DNA intensities then to intensities of the full-length mtDNA in each cell line at the starting time point and, in panel (**c**), additionally to the highest 2.1-kb fragment value on each blot. Error bars represent standard errors of the mean (SEM) in three independent experiments. Significance was calculated by applying one-way ANOVA test. *$P < 0.05$. **d, e** Cumulative relative frequencies (CRF) of ends around the 6910/6915 cutting site (represented by position 0) 2 h after mitoPstI induction as determined by single-molecule amplification of linker-ligated free mtDNA ends and subsequent Sanger sequencing. Positions indicated are relative to non-degraded ends at 6910 (**d**) or 6915 (**e**). CRFs were calculated from the following number of detected ends: **d** control (gray), $n = 36$; *MGME1* siRNA (red), $n = 39$; *POLG* siRNA (green), $n = 46$; *TWNK* siRNA (blue), $n = 37$; **e** control (gray), $n = 51$; *MGME1* siRNA (red), $n = 50$; *POLG* siRNA (green), $n = 37$; *TWNK* siRNA (blue), $n = 42$

(native vs. T4 polymerase treatment ratios of end counts normalized to total mitochondrial reads in diverse samples were 1.00 ± 0.13, SEM). If the majority of ends in the sample were not blunt, pre-treatment with T4 polymerase should have increased the number of ligatable ends dramatically. Since this was not the case, we concluded that the majority of non-degraded (Fig. 3b, c)

and partially degraded (Fig. 3d, e) mtDNA ends exists in vivo as blunt double strands, suggesting a simultaneous degradation of both mtDNA strands.

We observed prominent ends of partially degraded mtDNA at similar positions in both mitoEagI-expressing and mitoPstI-expressing control cells irrespective of mtRE used (Fig. 3d, e and

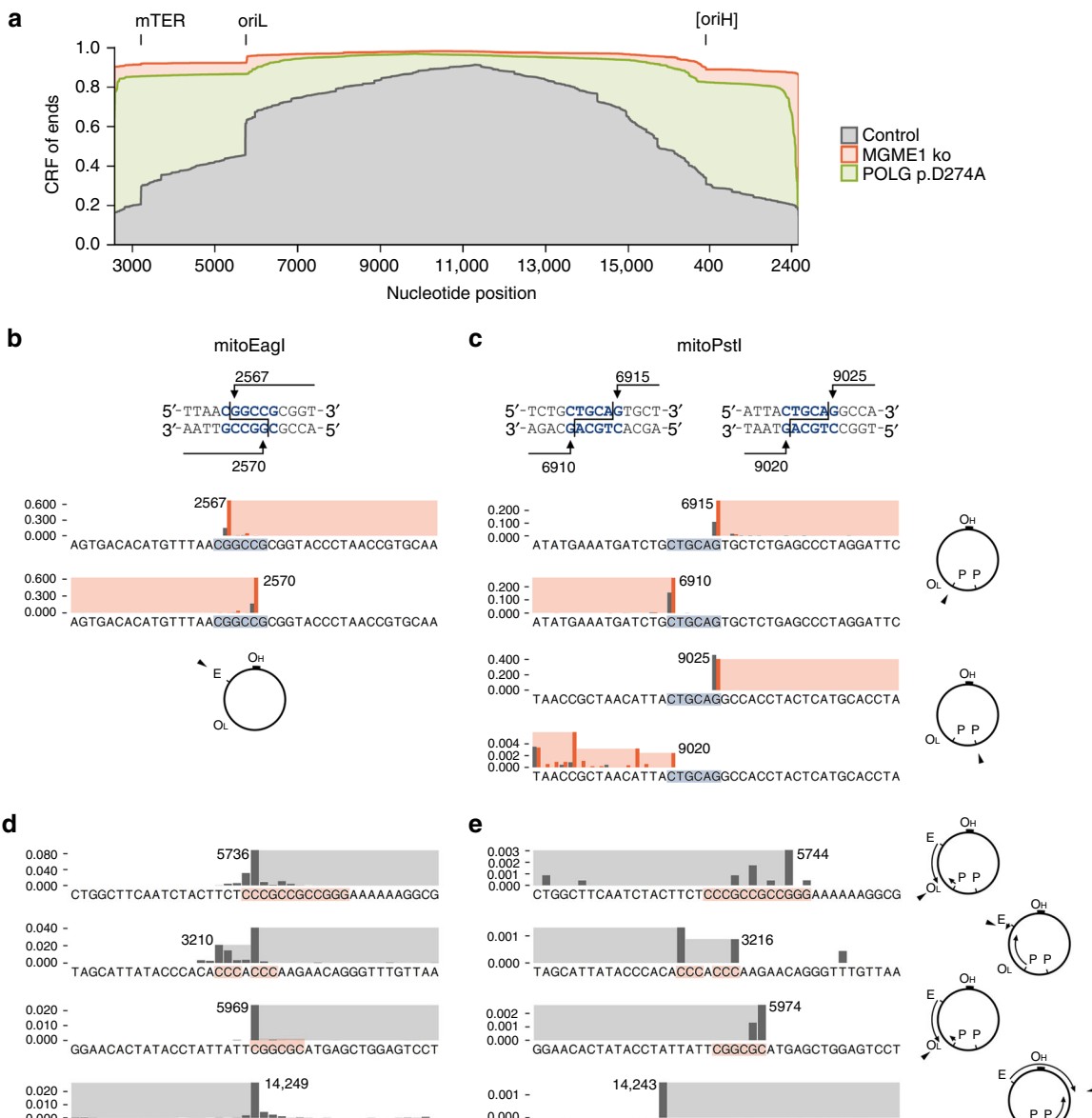

**Fig. 3** Free mtDNA ends in HEK 293 cells expressing mitoEagI or mitoPstI. **a** Cumulative relative frequencies (CRF) of mtDNA ends detected by ultra-deep sequencing of linker-ligated mtDNA 6 h after induction of mitoEagI expression. CRF values for both orientations of ends were combined into a single curve. MGME1 knockout (red) results in persistence of non-degraded ends. In POLG p.D274A knockin cells (green), over 80% of ends are detected within a distance of 600 base pairs from the cutting site. Functionally relevant sites associated with prominent clusters of ends are marked on the top (mTER, mitochondrial transcription termination site; oriL, replication origin for the light strand; [oriH], replication origin region for the heavy strand). **b**, **c** Relative frequencies of blunt mtDNA ends at the vicinity of cutting sites detected by ultra-deep mtDNA sequencing 6 h after induction of mitoEagI (**b**) and mitoPstI (**c**). Balk heights represent proportions of ends at specific nucleotide positions among all detected ends of the same orientation. Shadings indicate the retained part of mtDNA. Gray, control; red, *MGME1* knockout in mitoEagI and *MGME1* siRNA knockdown in mitoPstI. Blue shading indicates mitoEagI and mitoPstI recognition sites (schematically shown on the top). **d**, **e** Frequent mtDNA ends distal to the cutting sites as observed by ultra-deep mtDNA sequencing 6 h after induction of mitoEagI (**d**) or mitoPstI (**e**) in control cells. Shaded areas indicate the retained mtDNA fragment. Associated GC stretches are indicated by red shading (at least 3 consecutive Gs or Cs starting at less than 3 nucleotides difference from the end). Note that prominent ends are located at different sides of the same GC stretch depending on the main direction of degradation in mitoEagI-expressing and mitoPstI-expressing cells (indicated by arched arrows in the schemes on the side). The presence of non-degraded and selected partially degraded ends was confirmed by single-molecule PCR (Supplementary Fig. 5b)

Supplementary Fig. 5b). The most abundant end in mitoEagI cells was located at the replication origin region for the light strand of the mitochondrial genome (oriL). The position of the end corresponded to the second nucleotide of a 12 nucleotides long stretch of guanosines and cytidines (np5736, Fig. 3d). Ends in the mitoPstI model were clustering at the other end of the same GC stretch (np5744, Fig. 3e). Similarly, ends at one or the other side

of the same GC stretch were observed at a nearby position (np5969, Fig. 3d; np5974, Fig. 3e). The orientations of ends were the opposite at a GC stretch at the opposite side of the mitochondrial genome between positions 14,243 and 14,250 (np14,249, Fig. 3d; np14,243, Fig. 3e). This corresponded to the predicted main directions of degradation; the latter site is closer to the mitoEagI cutting site for clockwise degradation, and closer

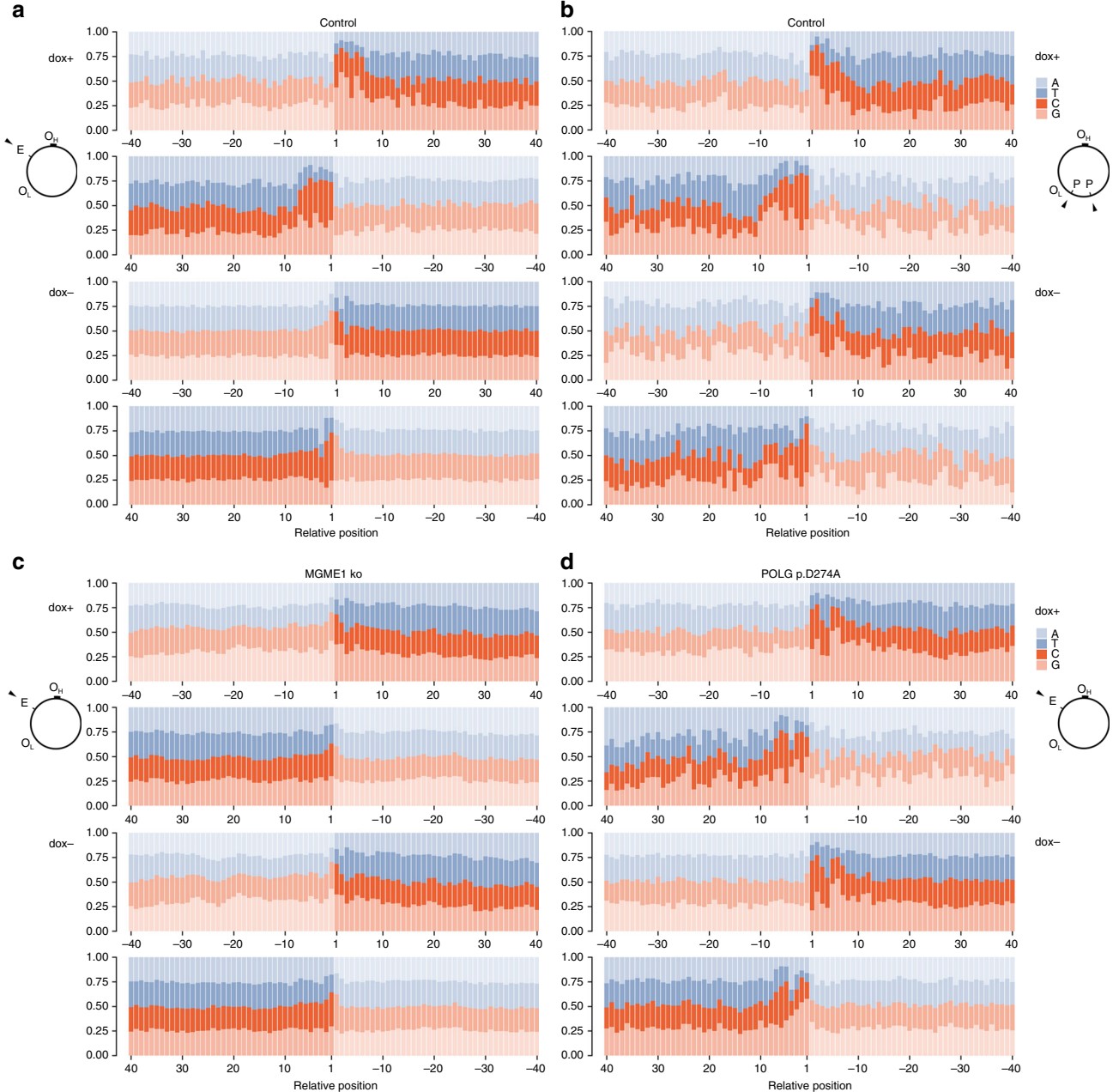

**Fig. 4** GC stretches decelerate mtDNA degradation. Relative frequencies of nucleotides at the vicinity of detected blunt double-stranded mtDNA ends are shown. Nucleotides surrounding all ends detected by ultra-deep sequencing of linker-ligated mtDNA were counted and their relative frequencies were normalized to overall nucleotide frequencies in the mitochondrial genome. Ends at the immediate vicinity of the cutting site were excluded from the analysis (mitoEagI: positions 2552–2585; mitoPstI: positions 6810–7015 and 8920–9125). To avoid bias through the most prominent ends in mitoEagI cells, corresponding positions were also excluded (positions 5732–5742 and 3208–3215). 'dox+', samples taken 6 h of inducing mitoEagI or mitoPstI expression. 'dox−', no induction, only leaky mitoEagI or mitoPstI expression. Upper set of panels shows ends generated by degradation in the forward direction, lower set of panels in reverse direction (according to reference numbering). Faded balks represent removed nucleotides. **a** Note that the high frequency of guanine and cytosine residues at the first 6 positions of linear mtDNA species upon induced cleavage in control mitoEagI cells. **b** Similar pattern can be observed in mitoPstI cells, although, cutting sites localize to different parts of the mitochondrial genome. **c** MGME1 knockout cells and **d** POLG p.D274A knockin cells do not show differences between induced and non-induced conditions. This is in line with the observation that newly generated ends do not undergo rapid degradation in the absence of MGME1 exonuclease or in the presence of exonuclease-deficient POLG

to the mitoPstI site for counterclockwise degradation (Fig. 3e, see schemes on the right-hand side). Examining overall nucleotide frequencies in the vicinity of all detected ends confirmed that ends of partially degraded mtDNA fragments tend, in general, to occur proximal to GC stretches depending on the direction of the degradation (Fig. 4a, b, dox+). This suggests that the removal of

nucleotides from the ends of linear mtDNA molecules is delayed by the presence of GC stretches. Notably, we did not observe an altered pattern of nucleotide frequencies upon mtRE induction in MGME1 knockout or mutated POLG cells (Fig. 4c, d). This confirms that newly generated linear molecules do not undergo rapid degradation in these cell lines.

**Ligation of mtDNA ends results in rearrangements**. It has been suggested that double-strand break repair is a key mechanism in generation of mtDNA deletions[23]. We, therefore, reasoned that, if linear mtDNA is not rapidly degraded, long-persisting linear mtDNA species might undergo double-strand break (end joining) repair which, in turn, would increase the probability of generating mtDNA rearrangements. Ultra-deep sequencing of mtDNA showed that the observed frequent breakpoints (Fig. 5a), occurring upon endonuclease-mediated linearization of mtDNA, corresponded to prominent blunt double-stranded mtDNA ends detected in mitoEagI-expressing cells (cf. Figure 3d). Similarly, rearranged mtDNA molecules carrying breakpoints corresponding to partially degraded ends of the 2.1-kb mtDNA fragment were detected in control mitoPstI-expressing cells, while breakpoints corresponding to non-degraded ends were present in MGME1 knockdown cells (Fig. 5b). This is suggestive that some of the mtRE-introduced breaks are re-ligated by a DNA end-joining activity operating in the human mitochondria. Corresponding breakpoints in control and MGME1 knockdown cells were also detected when screening for circular or linear concatemers of mtDNA fragments (Fig. 5c). To this end, knockdown of ligase III (LIG3), which has been proposed to play a role in mitochondrial non-homologous end joining[24], resulted in decreased abundance of concatemer structures (Fig. 5c).

It has been repeatedly reported that the majority (~85%) of breakpoints in patients' tissues carrying single mtDNA deletions or aging tissues accumulating multiple mtDNA deletions are flanked by short direct repeats[23]. To explain this phenomenon, replication slippage[25] or microhomology-mediated end joining[23, 26] of double-strand breaks were suggested as underlying mechanisms (Fig. 6b). In contrast, we observed a strongly reduced proportion of repeat-associated breakpoints in HEK 293 cells after mtDNA linearization (Table 1, Supplementary Table 2). Similar low proportions of breakpoints with direct repeats have also been reported in a mouse model expressing mitoPstI in skeletal muscle[27] or brain[28] (Table 1). Notably, reduced frequencies of repeat-associated breakpoints were observed for mtDNA multiple deletions in tissues of patients carrying pathogenic mutations in MGME1[8], POLG[8, 29] or TWNK[29] (Table 1, Supplementary Table 3). We therefore hypothesize that, in conditions where linear mtDNA persists due to insufficient degradation, non-homologous end joining of blunt mtDNA ends is more likely to generate mtDNA rearrangements that are not associated with direct repeats (Fig. 6b).

## Discussion

We show here that rapid degradation of linearized DNA, which is unique for the mitochondrial genome and surpasses double-strand break repair reactions, is executed by the same enzymes that are involved in mtDNA replication (Fig. 6a). This implicates novel, additional roles for the POLG, MGME1, and TWNK enzymes normally participating in mtDNA replication. Although the involvement of the MGME1 exonuclease in mtDNA degradation is not surprising, the exonuclease activity of the POLG polymerase has been discussed until now almost exclusively in the context of proofreading, i.e., the ability of the enzyme to remove mismatched nucleotides from the end of the newly synthesized DNA strand. Recently, however, it has been demonstrated that loss of POLG prevents purging of paternal mitochondrial DNA in fertilized eggs[6]. Similarly, it has been long known that the 3′-5′ exonuclease activity of T4 DNA polymerase is able to remove long stretches of one strand of double-stranded DNA in the absence of deoxyribonucleotide triphosphate (dNTP) substrates[30]. Our data suggest that POLG can perform a similar activity.

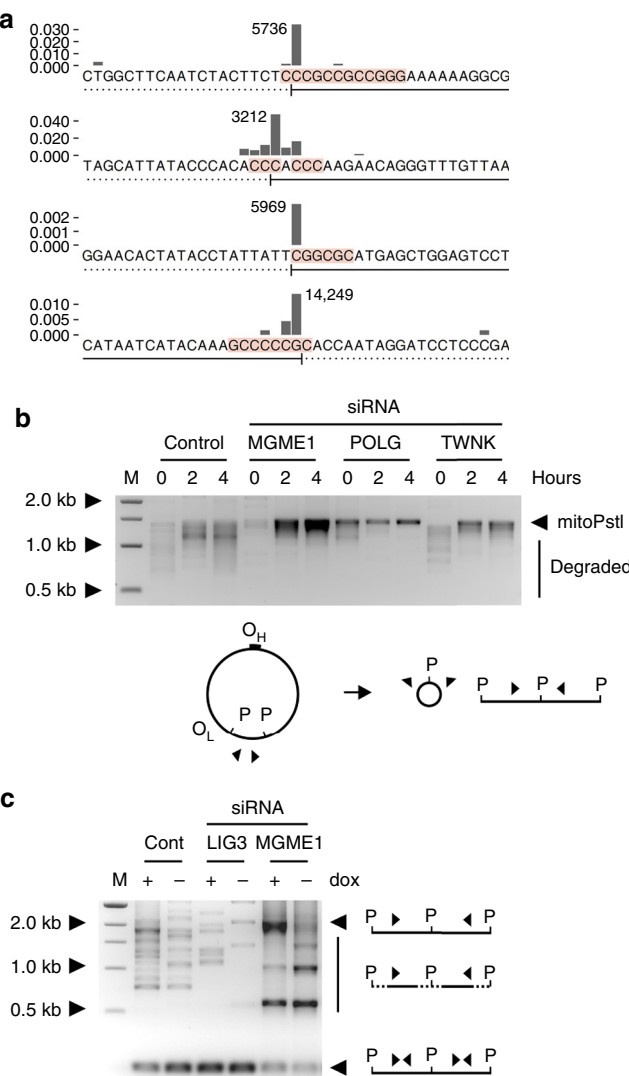

**Fig. 5** Rearranged mtDNA molecules in HEK 293 cells expressing mitoEagI or mitoPstI. **a** Positions of mtDNA breakpoints as detected by ultra-deep sequencing of mtDNA in control cells 6 h after induction of mitoEagI. Continuous lines represent retained parts of the mitochondrial genome. Dotted lines indicate deleted regions in rearranged mtDNA species. Note that the observed breakpoints correspond to frequent ends (cf. Figure 3d). **b** PCR detection of breakpoints corresponding to non-degraded and partially degraded ends of the 2.1-kb mtDNA fragment in mitoPstI-expressing cells. Amplification primers MT8282F and MT7682R (Supplementary Table 4) are shown in the scheme as arrowheads. Note that the majority of detected breakpoints correspond to partially degraded mtDNA ends in control cells, while MGME1, POLG, or TWNK siRNA treatments result in breakpoints mainly corresponding to non-degraded mtDNA ends. The exact positions of representative breakpoints determined by single-molecule PCR and sequencing are shown in Supplementary Table 2. **c** Concatemers of the 2.1-kb mtDNA fragment in mitoPstI-expressing cells as detected by long-extension PCR[8] using primers MT8194F and MT8387R (Supplementary Table 4). The shortest bands represent amplification products from unique copies of the 2.1-kb mtDNA fragment (lower arrowhead). Longer PCR products indicate the presence of multimers of the mtDNA fragment corresponding to non-degraded (upper arrowhead) or partially degraded (middle part) ends. Knockdown of the LIG3 gene decreases the abundance of concatemer species

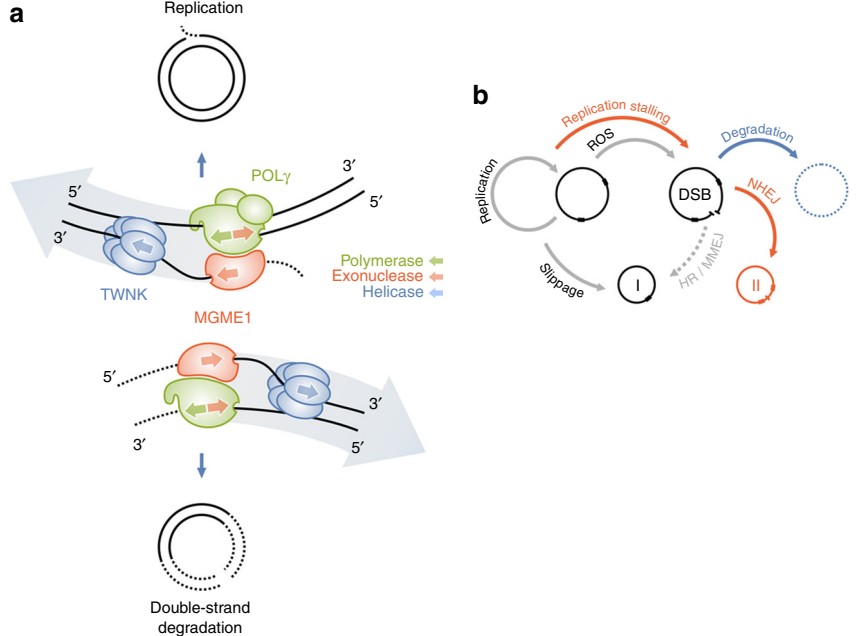

**Fig. 6** Models of double-strand degradation of mtDNA and its role in generation of rearrangements. **a** POLG, TWNK, and MGME1 play a role in both replication (upper half of the panel) and degradation (lower half of the panel) of mitochondrial DNA. Upon replication, the net movement of the complex (indicated by the large shaded arrow) corresponds to the polymerase activity of POLG. Under these conditions, MGME1 can remove flap structures, thus creating ligatable ends[9]. During degradation, net movement is reversed and corresponds to the exonuclease activity of POLG. **b** Proposed role of linear mtDNA degradation in generation of rearrangements. Since double-strand breaks (DBS) are efficiently removed in normal tissues (blue), most somatic mtDNA deletions are generated by replication slippage[25], and thus are typically associated with direct repeats (black boxes) around the breakpoints ('I', class I type of mtDNA deletions, Table 1). Repair by homologous recombination (HR) or microhomology-mediated end joining (MMEJ) is not an efficient pathway in animal mitochondria[34]. In the case of replication machinery dysfunction (red), frequent replication stalling leads to increased generation of double-strand breaks (DSB). Additionally, the breakdown of linear mtDNA is inhibited. The persistence of linear mtDNA favors the formation of class II deletions ('II') by ligase-III-dependent non-homologous end joining (NHEJ). Supporting this hypothesis, an increased frequency of class II mtDNA deletions (not being associated with direct repeats) is observed in patients carrying pathogenic mutations in *MGME1*, *TWNK*, and *POLG* (Table 1)

**Table 1 Low proportions of breakpoints with flanking direct repeats in mitoPstI-expressing cells and deficiencies of MGME1, POLG and TWNK**

| Sample | Class I[a] | Class II[a] | Significance[b] | Reference |
|---|---|---|---|---|
| mitoPstI-expressing HEK 293 cells | 24 | 57 | $7.2 \times 10^{-9}$ | Supplementary Table 2 |
| Skeletal muscle from mitoPstI mouse | 2 | 9 | $8.8 \times 10^{-4}$ | Srivastava & Moraes[27] |
| Neurons from mitoPstI mouse | 4 | 18 | $4.3 \times 10^{-6}$ | Fukui & Moraes[28] |
| MGME1 muscle (W152*, Y233C) | 42 | 39 | $7.9 \times 10^{-3}$ | Supplementary Table 3, Nicholls et al.[8] |
| TWNK muscle[c] (K319Q, M455T, R354P) | 15 | 31 | $8.2 \times 10^{-6}$ | Supplementary Table 3 |
| POLG muscle (W748S/L752P, R3P/A467T, Y955C, N468D/A1105T) | 59 | 81 | $2.7 \times 10^{-6}$ | Supplementary Table 3, Nicholls et al.[8], Wanrooij et al.[29] |
| Mitomap (single deletions in patients) | 92 | 38 | | www.mitomap.org[34] |

[a] Breakpoints of mtDNA rearrangements were categorized as Class I if they were associated with spanning perfect direct repeats of at least 5 nucleotides, otherwise Class II. Previously published data sets were re-analyzed with the same in-house Perl script that was used in this study (available upon request)
[b] P-values were calculated using Fisher's exact test comparing frequencies of mtDNA deletion classes in different samples to that of single deletions in patients as cataloged by Mitomap[35]
[c] TWNK-deficient patients whose tissues were used in this study were described previously[36]

A concerted action of POLG and MGME1 has been demonstrated for removing flap structures generated by POLG during replication[9]. Under this condition, the 5′-3′ exonuclease activity of MGME1 and the 3′-5′ exonuclease activity of POLG act on the same strand in opposite directions to transform flap structures into ligatable ends (Fig. 6a, upper part). Our data suggest that POLG and MGME1 can also act on complementary strands simultaneously progressing in the same direction (Fig. 6a, lower part). We demonstrated previously a physical contact between the two proteins[8]. Thus, their tight cooperation is probably not merely functional, but indeed physical.

Factors that turn on the degradation mode of the replication machinery are still to be identified. The switch between the opposite functions of the replication complex could be triggered by different types of template mtDNA (free double-stranded ends being degraded vs. 3′ ends at single-stranded stretches initiating replication). Additional protein factors might also play a role in alternative assembly of either a DNA replisome or degradosome. The fact that POLG2 is required for POLG to perform replication but is not essential for degradation raises the possibility that binding of POLG2 to POLG might be a relevant regulatory element. In analogy to the function of the T4 DNA polymerase,

acute mitochondrial dNTP depletion might also play a role in rapid mtDNA degradation after massive double-strand breaks.

Impaired mtDNA degradation might not only increase the abundance of abnormal linear mtDNA species[8, 31], but also promote the generation of multiple mtDNA rearrangements. The fact that the majority of breakpoints after mtRE-induced double-strand breaks were not associated with direct repeats suggests that rearranged mtDNA is generated by LIG3-mediated non-homologous end joining. Altogether, our findings provide evidence for a novel mechanism that broadens our understanding of linear mtDNA breakdown and the formation of mtDNA rearrangements.

## Methods

**Cellular models.** The sequences of the bacterial restriction endonucleases EagI and PstI were optimized for mammalian codon usage and fused with the mitochondrial targeting sequence of the human COX8B protein at the N-terminus and a hemagglutinin tag at the C-terminus and cloned into pcDNA5/FRT/TO (Fisher Scientific). MitoEagI and mitoPstI cell lines were generated in human HEK 293 T-REx cells (Fisher Scientific) allowing controlled expression by the tet-on system. To generate these knockin cell lines, HEK 293 T-REx cells were transfected with pcDNA5/FRT/TO/mitoEagI:HA (or pcDNA5/FRT/TO/mitoPstI:HA) and pOG44 at a molar ratio of 1:9 using Lipofectamine 2000 (Fisher Scientific). Successful integration was monitored by antibiotic selection with hygromycin B (100 µg ml$^{-1}$, Invivogen) and blasticidin (15 µg ml$^{-1}$, Invivogen). Single cell-derived colonies were tested for the expression of the mitoEagI and mitoPstI using an anti-HA antibody (Roche). MitoEagI cleaves the mitochondrial genome at a single site at position 2566/2570 creating 4-nucleotide 5′ overhangs (schemes in Figs. 1a and 3b). MitoPstI cuts at positions 6910/6914 and 9020/9024 generating two fragments of 2.1 kb and 14.5 kb with 4-nucleotide 3′ overhangs (schemes in Figs. 2a and 3c).

HEK 293 T-REx mitoEagI and mitoPstI cells were maintained in high glucose Dulbecco's Modified Eagle Medium (GlutaMAX™ DMEM, Gibco) supplemented with 10% tetracycline-free fetal calf serum (FCS, PAN Biotech), 10 U ml$^{-1}$ penicillin and 10 mg ml$^{-1}$ streptomycin (Gibco), 50 µg ml$^{-1}$ uridine (Sigma), 50 µg ml$^{-1}$ Hygromycin B (Sigma), and 15 µg ml$^{-1}$ Blasticidin S hydrochloride (Sigma). To induce expression of mitoEagI or mitoPstI, cells were administered with 20 ng ml$^{-1}$ doxycycline (Sigma) in fresh cell culture medium. Samples for DNA and RNA isolation were collected at 2 h intervals. Total DNA was isolated using QIAamp DNA Mini Kit (Qiagen). For RNA isolation, the cells were lysed in TRIzol Reagent (Invitrogen) and loaded on a QIAshredder (Qiagen). RNA was isolated using RNeasy Mini Kit (Qiagen). Inducible expression of mitoEagI and mitoPstI was confirmed by detecting increase in corresponding mtRE mRNA amounts by a factor of 56.8 ± 14.4 and 28.9 ± 2.6, respectively, using reverse transcriptase qPCR 6 h after induction. We additionally confirmed the inducible expression of mitoEagI and its mitochondrial localization by Western blotting (Supplementary Fig. 7), using a previously published protocol[32]. Efficient cleavage was confirmed by detecting rapid decrease of circular mtDNA in both models upon induction by Southern blotting (Figs. 1b and 2b). Cell lines were tested for mycoplasma contamination using LookOut Mycoplasma PCR Detection Kit (Sigma).

**Modified HEK 293 cells generated using CRISPR–Cas9.** The plasmid vector carrying single guide RNA (sgRNA) sequence for *MGME1* (5′-AGACCATTTG-CAGGCAGCTC-3′) and a separate plasmid coding for Cas9 were purchased from Origene. All other sgRNA plasmids were generated using the method described by Schmidt et al.[33]. Briefly, the BGH-pA-CMV-gRNA-Cas9 vector, carrying both sgRNA sequence and Cas9 gene, was linearized by ApaI/SpeI restriction endonuclease digestion; long single-stranded overhangs were created by T4 polymerase chew-back treatment in the presence of dTTP; and the vector was annealed with a universal reverse-strand oligonucleotide (CAS9UNIV) and a gene-specific sgRNA oligonucleotide (Supplementary Table 4) using the following conditions: temperature was gradually decreased from 75 °C to 60 °C in 37.5 min; 60 °C for 30 min; and 60 °C to 25 °C in 87.5 min. Annealed circular vectors were transformed to *E. coli* without ligation. A completely repaired, circular plasmid was produced in the bacterial host. Plasmids were isolated using Plasmid Miniprep Kit (Peqlab) and sequenced using CAS9PLF and CAS9PLR primers (Supplementary Table 4) to confirm the presence of the desired sgRNA sequence.

MitoEagI HEK 293 T-REx cells were transfected with CRISPR plasmids by plating the cells at a density of 10$^4$ cells per 96-well and performing transfection using the GeneJuice transfection reagent (Novagen) on the following day. To generate the *POLG* knockin, a 148-bp single-stranded oligonucleotide (POLG_HDRF; Supplementary Table 4) was additionally co-transfected. The oligonucleotide corresponded to the targeted region of the *POLG* gene but carried the desired missense point mutation, as well as silent point mutations for disruption of the protospacer adjacent motif (PAM) and a PstI restriction site.

Three days after transfection, cells were resuspended in 100 µl of fresh medium and counted. Cells suspensions were diluted and plated on 96-well plates at

calculated concentration of 0.5 cells per well. After 2–3 weeks of colony formation, single colonies were resuspended. Half of the cell suspension was re-plated on 96-well plates. Cells from the other half of the suspension were collected by centrifugation at 2000× *g* for 1 min and resuspended in 20 µl lysis buffer (1 mM CaCl$_2$, 3 mM MgCl$_2$, 1 mM EDTA, 1% Triton X-100, 10 mM Tris, pH 7.5) containing proteinase K (0.2 mg ml$^{-1}$, QIAGEN). Incubation at 65 °C for 10 min was followed by heat inactivation at 95 °C for 15 min. Lysates were directly used for PCR amplification and subsequent Sanger sequencing. In case of *POLG* knockin, PCR amplification products were screened for the loss of the PstI restriction site prior sequencing.

**Knockdown of genes by siRNA.** Cells were transfected with single or multiple Stealth RNAi™ siRNAs (Invitrogen) or Silencer Select siRNA (Ambion) for *ENDOG* at a final concentration of 30 nM using Lipofectamine® RNAiMAX transfection reagent (Invitrogen). Sequences of the used siRNA oligonucleotides are listed in Supplementary Table 1. Transfection was repeated after three days and followed by induction of endonuclease expression after six days. The efficacy of knockdown was determined at the mRNA level by reverse transcriptase qPCR using *PMPCA* as reference mitochondrial housekeeping gene (Supplementary Table 1) and, at the protein level, by Western blotting (Supplementary Fig. 6).

**Western blotting.** Cells were sonicated (2 × 15 s at 8% amplitude) in cell lysis buffer containing 20 mM Tris-HCl (pH 7.4), 150 mM NaCl, 1 mM EDTA, 1 mM EGTA, 1% Triton, and protease inhibitor cocktail (Roche 11836153001). Protein amount was determined using the BCA assay. Protein extracts (15–40 µg) in Laemmli buffer were resolved on a 10% polyacrylamide gel, and transferred to a Hybond™ P 0.45 PVDF-membrane (Amersham). Membranes were incubated overnight at 4 °C with the primary antibodies. The primary antibodies used for western blotting were as follows: β-actin (Gene Tex 124214, 1:10,000), MGME1 (Sigma HPA040913, 1:500), POLG (Santa Cruz SC-390634, 1:500), LIG3 (Sigma HPA006723, 1:500), TWNK (Sigma HPA002532, 1:1,000), FEN1 (Sigma HPA006748, 1:250), and ENDOG (Abcam ab9647, 1:1,000). Detection was performed with horseradish peroxidase-conjugated secondary antibodies (horse anti-mouse IgG HRP, Cell Signalling Technology 7076P2, 1:2,000; or goat anti-rabbit IgG–Peroxidase, Sigma A0545, 1:20,000) and SuperSignal West Pico Plus chemiluminescent substrate (Thermo Scientific) and signal was recorded on a ChemiDoc Imaging System (Bio-Rad).

**Southern blotting.** One microgram of total DNA was digested with BamHI restriction endonuclease (Fermentas). Products were separated on a 0.6% agarose gel at 40 V overnight along with DIG-labeled DNA Molecular Weight Marker II (Roche). The gels were alkaline treated and neutralized. DNA was blotted to Zeta-Probe membranes (Bio-Rad) and immobilized by baking at 80 °C for 30 min. Blots were hybridized with PCR-generated digoxigenin-labeled mitochondrial and nuclear (18S rRNA) probes (PCR DIG Probe Synthesis Kit, Roche; Supplementary Table 4) overnight at 48 °C. Chemiluminescent detection with anti-DIG-AP antibody F$_{ab}$ fragment (Roche) and CSPD (Roche) was performed and signal was recorded on a ChemiDoc Imaging System (Bio-Rad). If required, blots were stripped with 0.2 M NaOH + 0.1% SDS at 37 °C. Uncropped scans of the most important blots are shown in Supplementary Fig. 8.

**Quantitative PCR.** Quantitative PCR was used to determine the mtDNA copy numbers[8]. Primers 3922 F and 4036 R were used to amplify a minor arc segment of the mtDNA. Amplifications were performed on a MyiQ qPCR system (Bio-Rad, Munich, Germany) using 2 × SYBR Green qPCR Master Mix (Bimake) under the following conditions: 95 °C for 7 min and 45 cycles of 95 °C for 15 s and at a primer-specific annealing temperature for 1 min (Supplementary Table 4). Experiments were performed at three different concentrations of the template DNA, each concentration in triplicates. $C_T$ values were defined at the inflection points of fitted 4-parameter Chapman curves and were compared with those of the single-copy nuclear gene *KCNJ10* amplified by primers KIR835F and KIR903R.

Reverse transcriptase quantitative PCR was used to quantify mRNA amounts. A volume of 1 µg RNA was reverse transcribed with iScript Select cDNA Synthesis Kit (Bio-Rad) in a final volume of 20 µl. For qPCR amplification, primers spanning two exons were designed for each gene of interest (Supplementary Table 4). *PMPCA* was used as mitochondrial housekeeping reference gene.

**PCR detection of mtDNA rearrangements.** We investigated the presence of mtDNA rearrangements applying PCR-based techniques[8]. We used either long-range PCR that utilizes primers located on both sides of potential breakpoints (MT8282F and MT7682R; Supplementary Table 4) or long-extension PCR that is suitable to detect breakpoints associated with partially duplicated mtDNA molecules. In the latter case, primers would normally result in a small PCR product, but the long extension time enables the amplification of larger products if the primer-binding region is present more than once on the same molecule (primers MT8194F and MT8387R; Supplementary Table 4; scheme in Fig. 5c). PCR conditions were as follows: 95 °C for 2.5 min, 30 cycles of 92 °C for 20 s and 68 °C for 16 min, and finally 72 °C for 10 min.

**Ligation-mediated PCR**. Free ends of linear mtDNA molecules were detected by ligation-mediated PCR (LM-PCR)[8]. An asymmetric double-stranded oligonucleotide linker (Supplementary Fig. 5a) was generated by incubating primers LINK25F and LINK11R (Supplementary Table 4) at a concentration of 25 pmol μl$^{-1}$ each at 95 °C for 3 min and then gradually decreasing the temperature to 3 °C in 185 min. A volume of 0.2 μg of sample DNA with or without pre-treatment with T4 polymerase (Quick Blunting Kit, NEB) was ligated overnight with 100 pmol of the double-stranded linker using T4 DNA Ligase (NEB) at room temperature in a final volume of 60 μl. Amplification was done using an mtDNA-specific and a linker-specific (LINK25) primer, the latter at 1/8 concentration of the former. LM-PCR was performed under the following conditions: 95 °C for 2.5 min, and 30 cycles of 92 °C for 20 s and 68 °C for 1 min.

**Single-molecule PCR**. A single-molecule PCR approach was used to identify specific breakpoints or ends of mtDNA[8]. Briefly, single mtDNA molecules were amplified in 42 cycles of PCR using TaKaRa LA Taq Hot Start polymerase (Clontech; for large products when detecting breakpoints) or Ranger DNA polymerase (Bioline; for short products when detecting ends). Total template DNA was diluted to a grade at which only a part of multiple identical reactions resulted in amplification products (ideally less than 50%). Under these conditions, most of the positive reactions are likely to originate from a single mtDNA molecule. Amplification products representing single linker-ligated mtDNA ends were directly sequenced using an mtDNA-specific primer. Deletion breakpoints were mapped by re-amplifying single-deletion amplicons using diverse primer pairs located within the amplified region and direct sequencing of re-amplified products.

**Next-generation sequencing of linker-ligated isolated mtDNA**. $10^8$ HEK 293 cells 6 h after doxycycline induction or non-induced cells were used for isolation of mitochondria using magnetic beads coated with anti-TOM22 antibody (MACS, Miltenyi Biotec) and DNA was isolated using QIAamp DNA Mini Kit (Qiagen). A volume of 1.5 μg of purified DNA was ligated to the double-stranded linker as described above (Supplementary Fig. 5a) with or without pre-treatment with T4 polymerase and subsequently column purified with QIAamp DNA Mini Kit (Qiagen). Libraries were prepared and size selected by using the Illumina TruSeq nano DNA Sample Preparation Kit and Agencourt AMPure XP beads. One cycle of PCR followed to complete library adapter structure. Libraries were validated with the Agilent 2200 TapeStation and quantified by qPCR. Alternatively, the tagmentase-based NexteraXT protocol (Illumina) was tested. This technique, however, removes sequences from ends, thus reduces the probability to identify the linker sequence. 75-bp paired-end reads were generated using an Illumina HiSeq4000 instrument (Illumina, San Diego, CA, USA). For each sample, 0.7–1.4 × $10^7$ paired mitochondrial reads were obtained representing 20–50% of all reads and resulting in 1.2–2.3 × $10^5$ average coverage. Reads were aligned to sample-specific reference mitochondrial sequences and screened for the linker sequence using an in-house Perl script available upon request. Only full-length reads were considered where each nucleotide had a minimum quality score of 20. Approximately $10^5$ ends were identified in each direction in induced samples. Deletion breakpoints were identified in reads in which two parts of the sequence matched unambiguously two distinct regions of the mitochondrial genome, and none of the matches was shorter than 12 nucleotides.

**Data availability**. All relevant data are available from the authors. In-house Perl scripts used in the study are available upon request.

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

## Acknowledgements

This work was supported by grants of the Deutsche Forschungsgemeinschaft (KU 911/21-1, KU 911/21-2 to W.S.K. and ZS 99/3-1, ZS 99/3-2 to G.Z.) and BONFOR to V.P.; P. R.-G., P.A.G., M.J.S., and M.M. were supported by the Medical Research Council, UK (MC_U105697135). P. R.-G. was supported by "Fundação para a Ciência e a Tecnologia"

(PD/BD/105750/2014). The technical assistance of Susanne Beyer and Katlynn Carter (Bonn) is greatly acknowledged. We are thankful to Thomas Ebert (Institute for Molecular Medicine, University of Bonn) and Prof. Dr. Veit Hornung (Gene Center and Department of Biochemistry, Ludwig Maximilians University Munich) for providing us with the BGH-pA-CMV-gRNA-Cas9 vector.

## Author contributions

V.P., D.B., G.T., S.C., A.P.K., C.B., P.A.G., and J.A. conducted the experiments, M.J.S. and P. R.-G. constructed and characterized the HEK 293 T-REx cell models, and G.Z., M.M., and W.S.K. designed the experiments and wrote the paper.

## Additional information

**Competing interests:** The authors declare no competing interests.

