## [Peer Review File · Nature Communications]

Reviewers' Comments:

Reviewer #1:

Remarks to the Author:

In this manuscript by Peeva et al., the authors unravel the mechanism that leads to degradation of broken mtDNA. In this manuscript, the authors induce double-stranded breaks (DSBs) using mitochondrial targeted restriction enzymes that induce blunt DSB (EagI) and two sticky breaks (PstI) and show that mtDNA is rapidly degraded. Notably, degradation of damaged mtDNA is delayed/absent upon knockdown of MGME1, in the presence of mutant POLG, and following depletion of TWINKLE. On the other hand, knockdown of other exo/endonucleases including APEX2, FEN1, DNA2, EXOG, ENDOG, did not impact mtDNA degradation phenotype. Finally, the authors employ a novel sequencing approach to infer that DNA resection proceeds simultaneously on both strands.

The findings presented in this manuscript are novel, impactful, and timely. The study is potentially of interest to the readership of Nature Communication. However, the authors should address numerous experimental concerns prior to publication.

- 1- Figure 1. They present data from a single clone of MGME1 deleted cells. It is important to confirm these results using an independent clone that the authors have already generated.
- 2- Figure 1. The inducible Eag1 system was established for the first time in this manuscript, and it would be nice to show a western to monitor the expression and mitochondrial localization of the enzyme.
- 3- Figure 1a and 2a. In the representative gels for "control cells", distinct bands (~8Kb for Eag1 and ~1kb for Pst1) resist degradation. The authors must explain the appearance of these bands. Are they linked to OriL or OriH. Where they sequenced?
- 4- It is not clear as to how the reads in Figure 1b,c and Fig 2d,e were normalized. Depletion of MGME1 (and PolG mutant) is expected to lead to mtDNA depletion, which is not apparent from the presented graphs.
- 5- Mre11/CtIP are major end-joining nucleases that were previously detected in the mitochondria. It is rather important to address their effect on mtDNA degradation following Pst1 and Eag1 treatment.
- 6- Fig 2a. A 2.1Kb mtDNA fragment does not appear in the control sample, instead, a fuzzy band (2 hours) is replaced by a ~1kb fragment at 4 hour time point. Is the rate of degradation too fast, and they are missing it? If so, maybe perform Southern at 1 hour time point.
- 7- It is crucial to show westerns to monitor the RNAi mediated depletion of the various enzymes, including PolG, Twinkle, and MGME1.
- 8- The authors conclude that resection by PolG and MGME1 is tightly coordinated. It would be interesting (but not necessary) to address the mechanistic basis for the coupling of the two enzymes. Is the binding of PolG to DNA disrupted upon depletion of MGME1?
- 9- Figure 3. The authors conclude that non-homologous end joining of blunt-DNA ends is likely to generate infrequent rearrangements, regardless of direct repeats. It is crucial to address the mechanistic basis of this end-joining machinery. Is Ligase 3 involved in the formation of these rearrangements?
- 10- The authors should consider moving Supplemental Figure 4 to the main text. It is an important figure and could be highlighted in the manuscript.
- 11- The representation of Figure 3 (b-f) needs major modification. The figure is very confusing and poorly explained (both legend and text). It is difficult to keep track of the various junctions. It is not clear how they determined the frequency. What is their criteria for GC rich content? It is not clear how the authors reach the conclusion that the majority of ends are blunt.
- 12- Figure 3G. The authors monitor the re-circularization of PstI 2.1KB fragment as a proxy for end-joining activity. Southern blot using additional probes, is required to validate the appearance of rearranged molecules.
- 13- The models in figure 4b-c are overlapping and somehow confusing. The models can be merged and potentially additional colors (not just grey) can help connect the different pathways.

Reviewer #2:

Remarks to the Author:

The paper by Kunz's group has examined the consequences of double-stranded DNA breaks (DSBs) in mitochondrial DNA, and asked the question how the resulting linear mtDNA fragments are removed. Such linear fragments should not be able to replicate and have been reported to be eliminated quickly. The proteins involved in the elimination have remained unknown, and even the existence of DSB-repair mechanisms in mitochondria has recently been questioned (Moretton et al. Plos One 2017). This manuscript shows that components of the mtDNA replication machinery - Polg, Twinkle, and MGME1, but not other mitochondrial nucleases - are crucial for removal of linear mtDNA molecules. This discovery has important implications for understanding mechanisms of mtDNA homeostasis, as well as for therapeutic approaches aiming to selectively degrade disease-associated mtDNA molecules. The article proposes also an interesting model by which the core components of mtDNA replisome could function in two distinct modes, the replication and the degradation modes. The mechanism behind the switch between the two modes remains to be investigated.

The authors utilized HEK293 cells stably expressing (method how they were created is not described) mitochondrial-targeted endonucleases under inducible tet-promoter. These would introduce DSBs into mtDNA, if induced with tetracyclin. To these cell lines, they introduced either knockouts or knockin mutations to the genes of mtDNA processing enzymes, by Crispr/Cas9 system or shRNA. Further, they use deep DNA-sequencing to analyse the aberrant mtDNA ends generated in their different mutants. The paper is quite clearly written and the experimentation is of high quality.

Detailed comments:

1) In their abstract and discussion, the authors state that their finding of the roles of POLG, Twinkle and MGME1 in degradation of damaged mtDNAs is a novel mechanism to explain accumulation of abnormal mtDNA populations in patients with mtDNA maintenance disorders. Despite their nice mechanistic results, I find this statement over-generalization, as the authors study knockouts in cultured cells, and the patient mutations are not knockouts and manifest in postmitotic cells. To make the conclusion, the authors should at least test some patient mutations, and if they see them to replicate the findings of knockout lines, then the conclusion is better justified. Please modify the abstract accordingly and tune down the discussion concerning diseases.

2) A recent paper by Moretton et al. 2017 (mentioned but not at all discussed in the paper) found no evidence for the role of MGME1 in removal of linear mtDNA fragments, in a similar careful experimental system as reported here, using DSB induction by mito-targeted endonucleases. This is a clear conflicting result with the current paper and deserves appropriate discussion why this might be.

3) POLG-exo activity has been reported to be required for the MGME1 function in removal of replication-associated flaps, generated by POLG. The recent report by Uhler et al. (NAR 2016) shows this nicely, and also that MGME1 does not work on DS-templates. The similar effects of MGME1 knockout and POLG-exo-deficiency in this report fit to their concerted, even interdependent action in removal of DSB-generated linear fragments, and as both require ss-templates, the role of Twinkle is logical. This should be discussed, including the findings of Uhler et al, which is mentioned in the reference list, but not appearing in the text.

4) References (26-29) appear in the reference list but not in the text (26 and 27 appear only in the Table). Please update reference list to match the text.

5) Is POLG2 required for linear mtDNA elimination? ShRNA to take down POLG2 would be interesting.

6) Concerning the relevance of the findings for mtDNA maintenance diseases: exo-inactive mutations of POLG are not known in humans, whereas catalytic mutations (such as Y955C) are common, and also cause mtDNA rearrangements. Is the POLG catalytic activity required for DSB-handling? If the authors bring into their DSB-deficient cells a POLG with catalytic domain mutation and WT exo, can they rescue the defect of linear mtDNA elimination?

7) The potential role of MGME1 in recruiting Polg to the 'degradosome' could be considered. Do the authors find protein- or RNA-level induction of MGME1 upon DSB induction? Or change in its mitochondrial localization?

RESPONSE TO REFEREES

Reviewer #1 (Remarks to the Author):

In this manuscript by Peeva et al., the authors unravel the mechanism that leads to degradation of broken mtDNA. In this manuscript, the authors induce double-stranded breaks (DSBs) using mitochondrial targeted restriction enzymes that induce blunt DSB (EagI) and two sticky breaks (PstI) and show that mtDNA is rapidly degraded. Notably, degradation of damaged mtDNA is delayed/absent upon knockdown of MGME1, in the presence of mutant POLG, and following depletion of TWINKLE. On the other hand, knockdown of other exo/endonucleases including APEX2, FEN1, DNA2, EXOG, ENDOG, did not impact mtDNA degradation phenotype. Finally, the authors employ a novel sequencing approach to infer that DNA resection proceeds simultaneously on both strands.

The findings presented in this manuscript are novel, impactful, and timely. The study is potentially of interest to the readership of Nature Communication.

We would like to thank the referee for appreciating of our work.

However, the authors should address numerous experimental concerns prior to publication.

1- Figure 1. They present data from a single clone of MGME1 deleted cells. It is important to confirm these results using an independent clone that the authors have already generated.

Quantification of Southern blots (Figure 1b,c) included 3 independent experiments. Two of them were performed using p.I9Qfs*32, whereas the third one using the p.C10Sfs*33 MGME1 knockout clone. This is now clearly stated in the legend to Figure 1. Additionally, we introduced the new Supplementary Figure 2 displaying Southern blot and ultra-deep sequencing results for the p.C10Sfs*33 MGME1 knockout clone.

2- Figure 1. The inducible Eag1 system was established for the first time in this manuscript, and it would be nice to show a western to monitor the expression and mitochondrial localization of the enzyme.

Inducible expression and mitochondrial localization of the HA-tagged mitoEagI is now shown in the new Supplementary Figure 7.

3- Figure 1a and 2a. In the representative gels for "control cells", distinct bands (~8Kb for Eag1 and ~1kb for Pst1) resist degradation. The authors must explain the appearance of these bands. Are they linked to OriL or OriH. Where they sequenced?

These bands do not resist degradation in control cells, but are generated and degraded in a time-dependent manner with a delay that depends on the distance from the cutting site. This is compatible with a degradation process that starts at the free end and gradually progresses. This is now more clearly explained in the main text (page 3).

The free ends of these linear mtDNA degradation intermediates were extensively sequenced and are shown in Figure 3 for mitoEagI cells and Figure 2d,e for mitoPstI cells. The prominent band having one of the ends at the oriL region is now marked in Figure 1a and Supplementary Figs. 2a and 4a.

4- It is not clear as to how the reads in Figure 1b,c and Fig2d,e were normalized. Depletion of MGME1 (and PolG mutant) is expected to lead to mtDNA depletion, which is not apparent from the presented graphs.

Since Figures 1b,c and 2b,c aim to illustrate cutting efficiencies and the fate of linearized mtDNA molecules, we have not taken into account the differences in the steady-state mtDNA copy number. Therefore, we first normalized mtDNA band intensities to 18S rDNA intensities and then to the intensity of the full-length mtDNA band before induction ('0') in each cell line separately. In Figure 2c, 18S rDNA-normalized values were further normalized to the highest 2.1-kb band value on each blot. This is now more clearly explained in the legends to Figures 1 and 2.

Figure 2d,e shows cumulative relative frequencies (CRF) of ends, comparable to Figure 3a. In Figure 2d,e, the analysis was, however, restricted to a few hundred base pair distance from the cutting site and that is why it reaches 1 within this distance.

5- Mre11/CtIP are major end-joining nucleases that were previously detected in the mitochondria. It is rather important to address their effect on mtDNA degradation following Pst1 and Eag1 treatment.

Following Reviewer's comments, we performed siRNA-mediated knock down of *MRE11* and *RBBP8* (*CtIP*) in mitoEagl-expressing cells and included the corresponding Southern blots in Supplementary Figure 4a. We have not detected any effect of downregulation of these nucleases on mtDNA degradation following mitoEagl-mediated linearization.

6- Fig 2a. A 2.1Kb mtDNA fragment does not appear in the control sample, instead, a fuzzy band (2 hours) is replaced by a ~1kb fragment at 4 hour time point. Is the rate of degradation too fast, and they are missing it? If so, maybe perform Southern at 1 hour time point.

As shown in Figure 2b, most of the mtDNA is still circular at 2 hours after induction. Since even less efficient cutting is expected 1 hour after induction, we did not perform experiments at this time point. Notably, it requires two cleavage events to generate the 2.1-kb fragment. Given the observed rapid degradation of ends, it might be unlikely to detect mtDNA fragments of which the first cleaved site is still non-degraded at the time of the second cleavage. Nevertheless, a faint 2.1-kb band was more clearly detectable in other experiments (cf. Supplementary Figure 4b).

7- It is crucial to show westerns to monitor the RNAi mediated depletion of the various enzymes, including PolG, Twinkle, and MGME1.

In addition to the assessment of mRNA levels by RT-qPCR (Supplementary Table 1), we also performed Western blotting to demonstrate efficient knock-down in siRNA-treated cells at the protein level. These data are now shown in a new Supplementary Figure 6.

8- The authors conclude that resection by PolG and MGME1 is tightly coordinated. It would be interesting (but not necessary) to address the mechanistic basis for the coupling of the two enzyme. Is the binding of PolG to DNA disrupted upon depletion of MGME1?

Our present data cannot address the issue of whether the tight cooperation between POLG and MGME1 is only functional or these two proteins interact physically. Our previous data, however, indicate a physical contact between the two proteins (Nicholls et al. 2014, Hum Mol Genet. 23:6147-62, Figure 6). Since *in vitro* reconstruction of the minimal replication machinery did not include MGME1 (Korhonen et al. 2004, EMBO J. 23:2423–2429), it is definitely not required for POLG to bind to DNA. The arrangement of the two proteins during double-strand degradation has to be a subject of future investigations.

9- Figure 3. The authors conclude that non-homologous end joining of blunt-DNA ends is likely to generate infrequent rearrangements, regardless of direct repeats. It is crucial to address the mechanistic basis of this end-joining machinery. Is Ligase 3 involved in the formation of these rearrangements?

The referee raised an important point. Indeed, as indicated by recent literature, LIG3 is involved in the final sealing of DSBs during mitochondrial DNA end joining. Therefore, we monitored the generation of mtDNA rearrangements by investigating re-circularization / concatemerization of the small fragment in the mitoPstI model. We found that siRNA-mediated knock-down of LIG3 reduced the amount of rearranged mtDNA molecules in induced control mitoPstI cells. In the new Figure 5, we present these data (panel c) along with panels f and g of the original Figure 3.

10- The authors should consider moving Supplemental Figure 4 to the main text. It is an important figure and could be highlighted in the manuscript.

Supplementary Figure 4 has now been moved to the main body of the manuscript as the new Figure 4. Extended interpretation of these data has been introduced in the main text (page 7).

11- The representation of Figure 3 (b-f) needs major modification. The figure is very confusing and poorly explained (both legend and text). It is difficult to keep track of the various junctions.

To avoid confusion, we moved panels f and g from the original Figure 3 that contained the positions of mtDNA breakpoints corresponding to frequent ends detected by ultra-deep sequencing of mtDNA to the new Figure 5 (panels a and b). All the remaining panels in Figure 3 contain data exclusively on free mtDNA ends and not on junctions. The presented data are now explained in more detail in the main text (pages 6–7).

It is not clear how they determined the frequency.

Numbers on the y axis represent the proportion of ends at specific nucleotide positions among all detected ends of the same orientation. This is now more clearly stated in the legend to Figure 3.

What is their criteria for GC rich content?

Red-shaded nucleotides in panels d and e in Figure 3 simply indicate stretches of at least 3 G's and C's in the immediate vicinity (<3 nucleotide distance) of the detected ends. This is now more clearly stated in the legend.

It is not clear how the authors reach the conclusion that the majority of ends are blunt.

T4 polymerase treatment converts overhangs of DNA ends into blunt ends and does not affect ends that are already blunt *in vivo* (Supplementary Figure 5a). If the majority of the ends in the sample are not blunt, pre-treatment with T4 polymerase should increase the number of ligatable ends dramatically. In opposite to this, we observed that the number of detected ends (normalized to average coverage) was unaffected by T4 polymerase treatment. This suggests that most of the ends are blunt in the sample and therefore also ligatable without T4 polymerase treatment. This is now more thoroughly explained in the main text (page 6).

12- Figure 3G. The authors monitor the re-circularization of PstI 2.1KB fragment as a proxy for end-joining activity. Southern blot using additional probes, is required to validate the appearance of rearranged molecules.

The PCR method that was used in Figure 3g (now Figure 5b) is only able to identify ligated ends, but does not provide information on the structure of rearranged DNA molecules. In fact, long-extension PCR suggests that ligated molecules are linear or circular concatemers (duplications). These data are now shown in the new Figure 5c.

Our single-molecule quantification estimations indicate that, in our settings, the number of religated molecules is 2 magnitudes of order lower than that of free ends. Because of this, these heterogeneous religated molecules are not detectable by Southern blotting.

13- The models in figure 4b-c are overlapping and somehow confusing. The models can be merged and potentially additional colors (not just grey) can help connect the different pathways.

Panels b and c in Figure 4 (the new Figure 6) have been now merged to a single scheme and additional colors have been used.

Reviewer #2 (Remarks to the Author):

The paper by Kunz's group has examined the consequences of double-stranded DNA breaks (DSBs) in mitochondrial DNA, and asked the question how the resulting linear mtDNA fragments are removed. Such linear fragments should not be able to replicate and have been reported to be eliminated quickly. The proteins involved in the elimination have remained unknown, and even the existence of DSB-repair mechanisms in mitochondria has recently been questioned (Moretton et al. Plos One 2017). This manuscript shows that components of the mtDNA replication machinery - Polg, Twinkle, and MGME1, but not other mitochondrial nucleases - are crucial for removal of linear mtDNA molecules. This discovery has important implications for understanding mechanisms of mtDNA homeostasis, as well as for therapeutic approaches aiming to selectively degrade disease-associated mtDNA molecules. The article proposes also an interesting model by which the core components of mtDNA replisome could function in two distinct modes, the replication and the degradation modes. The mechanism behind the switch between the two modes remains to be investigated.

The authors utilized HEK293 cells stably expressing (method how they were created is not described) mitochondrial-targeted endonucleases under inducible tet-promoter.

The generation and characterization of the mitoEagI and mitoPstI cell lines is now provided in the corresponding Methods section and Supplementary Figure 7.

These would introduce DSBs into mtDNA, if induced with tetracyclin. To these cell lines, they introduced either knockouts or knockin mutations to the genes of mtDNA processing enzymes, by Crispr/Cas9 system or shRNA. Further, they use deep DNA-sequencing to analyse the aberrant mtDNA ends generated in their different mutants. The paper is quite clearly written and the experimentation is of high quality.

1) In their abstract and discussion, the authors state that their finding of the roles of POLG, Twinkle and MGME1 in degradation of damaged mtDNAs is a novel mechanism to explain accumulation of abnormal mtDNA populations in patients with mtDNA maintenance disorders. Despite their nice mechanistic results, I find this statement over-generalization, as the authors study knockouts in cultured cells, and the patient mutations are not knockouts and manifest in postmitotic cells. To make the conclusion, the authors should at least test some patient mutations, and if they see them to replicate the findings of knockout lines, then the conclusion is better justified. Please modify the abstract accordingly and tune down the discussion concerning diseases.

Discussion of the relevance of our findings has been refined taking into account the criticism of the Reviewer. The corresponding last sentence has been removed from the Abstract and the discussion has been modified accordingly.

2) A recent paper by Moretton et al. 2017 (mentioned but not at all discussed in the paper) found no evidence for the role of MGME1 in removal of linear mtDNA fragments, in a similar careful experimental system as reported here, using DSB induction by mito-targeted endonucleases. This is a clear conflicting result with the current paper and deserves appropriate discussion why this might be.

Discussion of the conflicting results by Moretton et al. has been introduced to the text (page 6). We identified some methodological differences between Moretton et al. 2017 and our knock-down approaches. Most importantly, Moretton et al. perform their experiments after 3 days of knock-down. Their and our Western blot data indicate that residual proteins are still present at this time point. We allowed a 6 days depletion time for proteins by repeating the siRNA-treatment after three days. In the new Supplementary Figure 6, we demonstrate that higher levels of protein depletion can be achieved after 6 days of siRNA treatment by Western blotting.

3) *POLG-exo activity has been reported to be required for the MGME1 function in removal of replication-associated flaps, generated by POLG. The recent report by Uhler et al. (NAR 2016) shows this nicely, and also that MGME1 does not work on DS-templates. The similar effects of MGME1 knockout and POLG-exo-deficiency in this report fit to their concerted, even interdependent action in removal of DSB-generated linear fragments, and as both require ss-templates, the role of Twinkle is logical. This should be discussed, including the findings of Uhler et al, which is mentioned in the reference list, but not appearing in the text.*

The model suggested by Uhler et al. for the concerted actions of MGME1 and POLG is now discussed in the text (page 9).

4) *References (26-29) appear in the reference list but not in the text (26 and 27 appear only in the Table). Please update reference list to match the text.*

References 28 and 29 were referred to in the legend to the original Figure 4, and now additionally in the extended discussion. The order of the references has been updated according to the guidelines of the Journal.

5) *Is POLG2 required for linear mtDNA elimination? ShRNA to take down POLG2 would be interesting.*

We performed siRNA-mediated knock-down of POLG2 in mitoEagl cells. These data indicate that POLG2 is not required for linear mtDNA elimination. These new data are now included in Supplementary Figure 4a and discussed in the main text (page 5).

6) *Concerning the relevance of the findings for mtDNA maintenance diseases: exo-inactive mutations of POLG are not known in humans, whereas catalytic mutations (such as Y955C) are common, and also cause mtDNA rearrangements. Is the POLG catalytic activity required for DSB-handling? If the authors bring into their DSB-deficient cells a POLG with catalytic domain mutation and WT exo, can they rescue the defect of linear mtDNA elimination?*

Unfortunately, we were not able to perform the suggested interesting rescue experiments due to the strict time restrictions for resubmission. Our novel data for POLG2 - which is required for catalytic activity of POLG, but apparently not required for degradation of linear mtDNA - allow to speculate that the catalytic activity of POLG might be also not required for elimination of linear mtDNA.

7) *The potential role of MGME1 in recruiting Polg to the 'degradosome' could be considered. Do the authors find protein- or RNA-level induction of MGME1 upon DSB induction? Or change in its mitochondrial localization?*

Although, our previous data indicate a physical contact between POLG and MGME1, it is not clear whether such physical interaction is required to explain the observed results. Also, POLG in the 'degradosome' might be simply a reversed replicating POLG, thus, without a need of a degradation-specific recruiting. These considerations are now included in the extended discussion (page 9).

We did not observe any up-regulation of MGME1 either at mRNA or at protein level under conditions when double-strand breaks had been induced. Corresponding data are now shown in a new Supplementary Figure 3 and discussed in the text (page 4).

Reviewers' Comments:

Reviewer #1:

Remarks to the Author:

The authors addressed my concerns.

Reviewer #2:

Remarks to the Author:

The authors have responded sufficiently to my comments with additional high-quality experimentation. I have no further comments.

Response to the REVIEWERS' COMMENTS:

Reviewer #1 (Remarks to the Author):

The authors addressed my concerns.

Reviewer #2 (Remarks to the Author):

The authors have responded sufficiently to my comments with additional high-quality experimentation. I have no further comments.

We would like to thank the reviewers for the helpful and constructive comments which helped to improve the manuscript.